# Learning by Teaching: Engaging Students as Instructors of Large Language Models in Computer Science Education

**Xinming Yang**
The Graduate Center, CUNY
New York, NY, USA
xyang1@gradcenter.cuny.edu

**Haasil Pujara**
Hunter College, CUNY
New York, NY, USA
haasil.pujara40@myhunter.cuny.edu

**Jun Li**
Queens College & The Graduate Center, CUNY
New York, NY, USA
jun.li@qc.cuny.edu

## Abstract

While Large Language Models (LLMs) are often used as virtual tutors in computer science (CS) education, this approach can foster passive learning and over-reliance. This paper presents a novel pedagogical paradigm that inverts this model: students act as instructors who must teach an LLM to solve problems. To facilitate this, we developed strategies for designing questions with engineered knowledge gaps that only a student can bridge, and we introduce Socrates, a system for deploying this method with minimal overhead. We evaluated our approach in an undergraduate course and found that this active-learning method led to statistically significant improvements in student performance compared to historical cohorts. Our work demonstrates a practical, cost-effective framework for using LLMs to deepen student engagement and mastery.

## 1 Introduction

Large language models (LLMs) represent a significant advancement in artificial intelligence. After rapid developments, LLMs have demonstrated impressive improvements in natural language processing (NLP) tasks (Zhao et al., 2023), surpassing traditional models in areas such as task generation, translation, question answering, and code generation (Yang et al., 2024; Cloutier & Japkowicz, 2023). Trained on massive datasets, LLMs are able to generate human-like text. Some of the most popular LLMs include GPT models (Kalyan, 2023; OpenAI et al., 2023) developed by OpenAI, PaLM (Anil et al., 2023), and Gemini (Team et al., 2023; Gemini Team Google et al., 2024) from Google, Claude models (Anthropic, 2024) from Anthropic, and LLaMa models (Touvron et al., 2023a) from Meta.

The significant performance of LLMs has prompted their exploration in computer science (CS) education (Raihan et al., 2025). Here, a crucial distinction exists between using LLMs as a utility (*e.g.,* for professional code generation) and as a tutor for foundational learning. Our work focuses on the latter. Within this pedagogical context, the dominant paradigm casts the LLM as a virtual tutor, *i.e.,* an assistant that explains concepts, debugs code, and answers questions (Denny et al., 2024; Liu & M'Hiri, 2024; Kazemitabaar et al., 2024). However, this model carries significant risks. Major concerns include the potential for academic dishonesty (Perkins, 2023) and, more critically for learning outcomes, that students may become overly dependent on the LLM for reasoning instead of developing their own understanding (Wang et al., 2024; Abdelghani et al., 2023; Bastani et al., 2024). This can foster passive learning and hinder the development of practical skills.

In this paper, we argue that the existing efforts on applying LLMs in education use LLMs as an assistant which replace some necessary effort a student needs to make during learning. To prevent this negative effect, we propose to use LLMs in a reversed role, *i.e.,* let a student

teach an LLM to solve a question. Therefore, a student will be pushed to understand the materials behind the given question. This approach leverages the well-documented protégé effect, where the act of teaching material deepens the instructor's own understanding and mastery (Roscoe & Chi, 2007; Chase et al., 2009).

To achieve this objective, dedicated questions need to be designed as many existing course questions can already be solved by LLMs at a human level. Our objective is to make the questions hard enough such that they cannot be solved by an LLM, but can be solved with appropriately designed prompts, which is the expected answer from students. As it is non-trivial to find questions at the appropriate level of difficulty, we develop a series of strategies for designing such questions.

To equip students to solve these challenging problems, our methodology incorporates techniques for both student guidance and system-level validation. Students are guided to structure their solutions using established prompt engineering principles. They use Chain-of-Thought (CoT) (Wei et al., 2022; Kojima et al., 2022) to deconstruct the problem into a step-by-step process, often in combination with Few-Shot Prompting (Brown et al., 2020) to provide complete, reasoned examples as templates for the LLM. To then robustly evaluate this student-generated instruction, our Socrates system employs the Self-Consistency inference strategy (Wang et al., 2022; Min et al., 2023), which sends the student's prompt multiple times to ensure the proposed solution is consistently effective.

To facilitate the deployment of the above designs, we implement Socrates, a system that provides a playground for students to engage with our specially designed questions and a grader that uses an LLM for verification. The system is designed for ease of use by instructors, requiring minimal programming knowledge. To validate the effectiveness and viability of this novel paradigm, this paper addresses the following research questions:

1. To what extent does our "learning by teaching" approach impact student performance compared to traditional methods in an undergraduate computer science course?

2. How effectively can students guide an LLM to correctly solve complex, custom-designed problems, and what are the associated computational costs and practical considerations of our framework?

We evaluate these questions through a deployment in an undergraduate course at CUNY Queens College. Our findings demonstrate that this approach yields statistically significant improvements in student outcomes at a low operational cost, validating both its pedagogical value and its practical feasibility for broader adoption.

## 2 Background

Recent generative LLMs, such as OpenAI's GPT series (Kalyan, 2023; OpenAI et al., 2023), Meta's LLaMa series (Touvron et al., 2023a;b; Meta AI, 2024), and Google's Gemini (Team et al., 2023), have demonstrated remarkable capabilities in complex reasoning and code generation. Such capabilities are not static but are elicited through carefully crafted instructions provided in context, *i.e.,* prompt engineering. This reliance on prompting makes them highly suitable for our pedagogical paradigm, which centers on the student's ability to formulate effective instructions. Hence, our pedagogical framework is built upon established prompt engineering principles designed to elicit more robust and logical outputs from LLMs.

**Chain-of-Thought (CoT):** The CoT principle (Wei et al., 2022; Kojima et al., 2022) guides an LLM to produce intermediate reasoning steps or "thoughts" before the final answer, mimicking how humans break down complex problems (Zhou et al., 2023). By doing so, it encourages the model to generate more structured and logical responses, especially in tasks that require reasoning (Chu et al., 2024; Wei et al., 2022). For example, given an arithmetic word problem, the model would generate step-by-step reasoning like "There are 3 cars initially. 2 more cars arrive. So $3 + 2 = 5$ cars in total." before stating the final answer of 5 cars. CoT has been shown to significantly improve performance on complex tasks. We leverage this by designing assignments that require students to provide these explicit, step-by-step instructions, thereby teaching the LLM to solve the problem logically.

**Few-shot prompting.** Another cornerstone of our approach is few-shot prompting, a technique where providing a model with a small number of solved examples in its context dramatically improves its performance on similar, unseen problems (Brown et al., 2020). Instead of simply using this for model performance, we leverage it as a pedagogical instrument. In our framework, students are required to construct these few-shot examples, often including not just the final answer but also the intermediate reasoning steps (a combination of few-shot and CoT prompting). This process forces students to deconstruct a problem, identify its core patterns, and articulate a generalizable solution, thereby reinforcing their own understanding before they "teach" it to the LLM.

While the prompt engineering principles above help students formulate clearer instructions, they do not eliminate the inherent stochasticity of LLM outputs. A well-designed prompt can still produce an incorrect result due to the probabilistic nature of the model. A more robust result can be obtained by an inference-time decoding strategy, such as self-consistency (Wang et al., 2022). Instead of relying on a single greedy decoding of the reasoning path, it samples multiple diverse reasoning paths from the language model and selects the most consistent or reliable final answer by marginalizing all sampled reasoning paths (Bartsch et al., 2023; Min et al., 2023). The intuition is that for complex reasoning problems, multiple valid ways of thinking can lead to the same correct answer. By considering the consensus across many reasoning paths, self-consistency can boost accuracy compared to just taking the greedy output (Chen et al., 2023). In our Socrates system, this is implemented by allowing students to query the LLM multiple times and using a majority consensus to determine success, making the interaction more robust against single-run failures.

## 3 Related Works

In computer science education, LLMs are most commonly employed as virtual teaching assistants. Numerous studies have explored the benefits and challenges of this paradigm, using models to provide instructional support, generate and grade questions, and enhance accessibility in both traditional and online learning environments (Markel et al., 2023; Denny et al., 2024; Liu & M'Hiri, 2024; Kazemitabaar et al., 2024). For example, Liu et al. (2024) discuss the pivotal role of LLMs in CS education by integrating AI-driven instructional supports in courses like Harvard's CS50. Moreover, the work by Liu & M'Hiri (2024) exemplifies the focus on refining this tutor model. Their primary contribution was investigating how to structure interactions with a GPT-3.5 assistant to provide accessible programming help while implementing safeguards to maintain academic integrity. Similarly, the application of LLMs in economically disadvantaged educational settings is further explored by Choi et al. (2023), who investigate the deployment of AI tools in areas with limited educational infrastructure. Their findings indicate that, despite challenges, LLMs can significantly improve educational delivery and accessibility.

The dynamics of student engagement with LLMs are examined by Abdelghani et al. (2023), who question whether students can remain active learners in environments enriched with AI tools. Their study highlights the delicate balance required to maintain student interaction without fostering over-reliance on AI technologies. The authors present a nuanced analysis that highlights the potential risks associated with integrating generative AI tools into the learning environment. They argue that while LLMs can provide substantial educational benefits, there is a significant risk that students may become overly reliant on these AI systems. This over-reliance, which is also proposed and confirmed by Wang and Xu, could potentially diminish the students' motivation and ability to engage deeply with the learning material on their own (Wang et al., 2024). Their findings suggest that without careful implementation and continuous monitoring, the introduction of LLMs into the classroom could inadvertently undermine the very educational engagement they aim to enhance. This is echoed by Jeon & Lee (2023), who caution that without adequate pedagogical framing, educators may inadvertently misuse LLMs, *e.g.,* by providing overly-scaffolded prompts that allow students to retrieve answers without engaging in productive struggle, thereby undermining the learning process. This concern of fostering over-reliance is empirically supported by Bastani et al. (2024) in a study with approximately 1000 high school students (Grade 9-11) in mathematics. They found that while initial access to GPT-4 boosted perfor-

mance, students who had the tool removed later performed worse than those who never had access. This finding highlights the inherent risk of using LLMs as assistive tools and underscores the need for alternative pedagogical models, like ours, that aim to prevent such detrimental dependencies.

Collectively, these studies form a comprehensive view of the nuanced implications of LLMs in educational settings, suggesting the need for further research and development in terms of the interaction between LLMs and students to maximize the potential of LLM-enhanced education (Markel et al., 2023). In this paper, we present our novel way of utilizing LLMs in CS education by reversing the role of the LLM as a student. Our approach is grounded in the well-established protégé effect, which demonstrates that the act of preparing to teach, explaining concepts, and resolving confusion for another agent forces the "teacher" to organize their own knowledge more effectively and identify gaps in their understanding (Chase et al., 2009; Roscoe & Chi, 2007). By positioning the student as an instructor who must teach an LLM, we aim to transform learning from passive information consumption into a process of active knowledge construction, directly targeting the issues of engagement and over-reliance identified in the literature.

## 4 Methodology

Our methodology is designed to create a pedagogical interaction where the student must actively construct and articulate their knowledge to instruct an LLM. This is achieved through two complementary components: the design of the problems themselves, and the techniques used to guide the student's instruction process.

### 4.1 Question Design

The success of our role-reversal pedagogy hinges on designing questions that LLMs cannot solve independently, thereby preventing solution retrieval from training data and forcing a guided reasoning process. The pedagogical value of this process is grounded in established learning theories. Our strategy of creating novel, underspecified scenarios is a direct application of constructivist learning theory, where students build a more rigorous understanding by defining the rules of a new system (Harel & Papert, 1991; Kafai & Resnick, 1996). This approach is fundamentally different from merely prompting an LLM to "act like a student of level X." Such simulation can be inconsistent and does not create a genuine information dependency (Mannekote et al., 2025; Kumar et al., 2025). By engineering a genuine knowledge dependency, we create a robust pedagogical setup where the student's role as instructor is essential. This section introduces the two core strategies designed to implement this pedagogically-grounded approach. We applied these strategies to the modules of an undergraduate course. A detailed description of this application is provided in Appendix A, and full example questions are available in Appendix B.

**Strategy 1: Creating Non-Existing Scenarios.** This strategy embeds known concepts within a completely novel context defined by arbitrary, multi-part rules that an LLM cannot infer from its training data. For instance, a question in data representation might define a signed number system where digits A and B represent 0 and 1, and the number's sign is determined by its case: an all-uppercase string like BABA is positive, while an all-lowercase one like baba is its negative equivalent. This two-part rule (digit mapping plus a case-based sign convention) is outside any standard system and requires explicit definition by the student. Similarly, in assembly language, a task could require the student to define and use a hypothetical instruction like SWAPADD R1, R2, R3, which first swaps the values in registers R1 and R2, and then adds the new values, storing the result in R3. The procedural and arbitrary nature of this multi-step operation makes it impossible for an LLM to guess its function, forcing it to rely entirely on the student's explanation.

**Strategy 2: Involving Guided Mathematical Reasoning.** This strategy leverages complex, multi-step procedures where LLMs are known to struggle without explicit step-by-step guidance (Wei et al., 2022; Kojima et al., 2022). The student's task is not to provide the final answer but to act as a guide for the entire logical process. For example, to convert a Boolean

function like $y = a(b + bc')$ into its canonical sum-of-minterms form, the student must guide the LLM through the full algorithm. The student would first instruct the model to "apply the distributive law to get $ab + abc'$", then to "identify the missing variable $c$ in the term $ab$", followed by instructing it to "expand the term $ab$ by ANDing it with $(c + c')$", and finally to "remove the duplicate minterm $abc'$". Each command corresponds to a discrete step in the algorithm, compelling the student to master the procedure itself, not just the final outcome. Meanwhile, this strategy focuses on high-level logic over rigid syntax, significantly flattening the learning curve and allowing students to concentrate on core concepts without the initial barrier of mastering a formal language.

## 4.2 Guiding Student Instruction

Our pedagogical approach requires that the LLM cannot solve the problem independently, creating a knowledge gap for the student to fill. This makes the use of less-capable, and thus more affordable, LLMs not only feasible but preferable. However, this choice necessitates that students learn how to effectively guide the model, which can be a challenge. Students can be frustrated when trying different prompts with instructions they think are sufficient, with no correct output from the LLM. Therefore, we provide guidance based on established methods for structuring student instruction and ensuring its robustness. Requiring students to provide step-by-step guidance aligns with problem decomposition, a core component of computational thinking that reinforces procedural thinking (Brennan & Resnick, 2012; Lye & Koh, 2014).

**Guiding Reasoning with Chain-of-Thought.** We apply the CoT principle (Kojima et al., 2022; Wei et al., 2022) by guiding students to structure their answers as a series of simple, sequential steps for the LLM to follow. For all questions in general, we make students aware of the CoT principle and encourage them to decompose the work in their solutions into steps and make each step simple enough for the LLM to follow. For some more difficult questions which an LLM achieves a much lower success rate than others with answers from volunteer students during testing, we split the answer into steps, and required students to give the answer to each step. Eventually, the answers to all steps will be combined into the prompt sent to the LLM, enforcing the application of CoT. Besides, we apply simple yet efficient tricks in CoT, *e.g.,* by including a phrase "let's think step by step" in the prompt.

**Demonstrating with Few-Shot Examples.** We leverage few-shot prompting (Brown et al., 2020) by requiring students to provide several examples of test cases with corresponding input-output pairs. This serves a dual purpose: it provides a clear output format for the LLM while compelling students to generalize the problem by constructing their own valid examples. Few-shot learning can also be integrated with CoT to further enhance the quality of the LLM's output. Hence, besides asking students to give examples of test cases, we also ask students to give detailed, step-by-step solutions to such test cases. These examples serve as a guide for the LLM, setting the framework of how to approach similar problems. When faced with a new problem, the LLM uses the few-shot examples as a reference to structure its chain-of-thought, sequentially working through the problem while aligning its approach with the demonstrated examples. For students, this integration gives them a chance to solve the problem from simple examples, which can then be extended to general cases.

**Ensuring Robustness via Self-Consistency.** To manage the inherent stochasticity of LLM outputs, we apply the principle of self-consistency (Wang et al., 2022). In our system, a student's prompt is sent to the LLM multiple times. This has direct pedagogical value: when a student's instructions fail on some trials but succeed on others, it helps them distinguish between LLM variability and a fundamental flaw in their own logic. Conversely, consistent failure across all trials provides a strong signal that their instructions must be refined. This process helps students diagnose ambiguity in their prompts and formulate more robust instructions (Madaan et al., 2023). The student's answer is considered successful only if the number of correct outputs meets a predefined threshold.

# 5 System Design

To support these designs, a system is needed for both students and instructors, with its requirements grounded in established pedagogical principles. For students, the system must provide a controlled and uniform environment to ensure assessment fairness and equity for all learners (Association et al., 2014). The interaction environment must also be non-modifiable to uphold academic integrity, a key concern when integrating generative AI (Denny et al., 2024). For instructors, particularly those with limited programming knowledge, the system should be easily deployable. While a full human-computer interaction (HCI) evaluation is beyond the scope of this paper, our design focuses on these core pedagogical requirements, a common approach when the primary goal is to evaluate learning outcomes (Koedinger et al., 2013).

We now present a system called Socrates that incorporates LLMs as virtual students to enhance learning in CS courses, satisfying above requirements. Fig. 1 illustrates the pedagogical workflow enabled by the system. Our core contribution lies in this novel interaction model rather than in a new underlying technical architecture. An instructor can design an assignment file with questions, which can be sent to the playground and the grader. The playground provides a web-based UI generated from the assignment file, and lets students provide answers and interact with LLMs. The answers from a student for an assignment can be recorded into an answer file, which will be sent to the grader for grading.

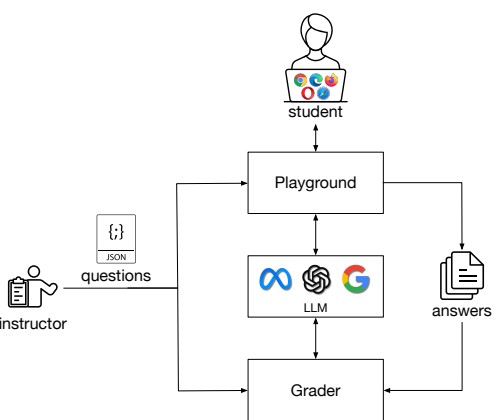

Figure 1: The architecture of Socrates.

**Questions.** In Socrates, an instructor can define assignment questions in a text file in JSON format. We chose JSON for its flexibility and ease of use in a research prototype context, allowing for rapid development and modification of question structures, and a graphical user interface (GUI) for question design is a direction for future work. Each question includes a description field detailing the scenario and requirements for students. It can also have one or more answer areas, allowing students to provide answer(s) in the playground. Typically, a prompt combining the question description with the student's answer(s) will be sent to an LLM. Instructors can also specify the LLM used for a particular question and an additional prompt, such as a test case or simply "let's think step by step". Multiple test cases may be provided for students to select in the playground, with the LLM solving the question accordingly. Each question may also be associated with a few more fields for the grader only, such as dedicated test cases, their sample correct output, and the threshold for self-consistency.

To help students get familiar with the format and requirements of the assignment, the first question may be set to be a demonstration with a sample answer given such that students can simply watch the output from the LLM. The instructor may also specify in the question the number of times the prompt will be sent to the LLM for self-consistency in the playground and grader. In this case, we allow some question(s), especially the first one, to be specified as a demo in the assignment file, with an additional field for the sample answer inside this question.

Besides questions, an assignment file may also contain an overview of the assignment and passcodes for all students. The overview can provide a general introduction to the whole assignment to students before all the questions. A passcode is a unique string assigned to each student and should be distributed to each student separately, which can be used to match the current user to a specific student in the playground and prevent unauthorized usage of LLMs.

**Playground.** The playground in Socrates provides an interactive environment for students to work on questions designed by an instructor. It first takes the assignment file that we

described above as its input. Thanks to its format in JSON, we can easily convert it into a Jupyter notebook where different fields of a question will become a text cell or a code cell with one or multiple widgets for taking input from students. The notebook also includes Python code and widgets for verifying the student's passcode and submitting the prompt to the LLM, and displaying the output from the LLM. At the end of the notebook, there will also be a button widget allowing students to submit their final answers. The submission will be saved as an answer file on the server running the playground, which can be sent to the grader for grading offline.

After conversion, the playground launches a Voilá (The Voila Development Team, 2024) session that turns the Jupyter notebook into an interactive web UI. The student can then get access to the assignment in a web browser. Hosting the Jupyter Notebook in the Voilá session hides the code from students, making them only work with questions without seeing the necessary details. It makes students unable to modify the code, making sure they work in a controlled and uniform environment. It also prevents students from getting sensitive data such as the API key of the LLM.

**Grader.** The grader is a component for the instructor to grade students' submissions. The grader reads the question file to get the information about the questions and then grades each answer file from the students. When grading each question, the same prompt is generated in the grade as in the playground and sent to the LLM for the output. Multiple requests with the same prompt may be sent to the LLM as specified in the question file. An instructor may specify additional test cases for grading only in the question file.

After getting the output from the LLM, the grader sends the output for verification, during which the grader uses an LLM to verify if the output is correct by comparing it with the sample's correct output. The prompt instructs the LLM to give either Yes or No to indicate if the output is correct compared to the sample's correct output. The answer is correct if the number of Yes is no less than the threshold of self-consistency, such as when the LLM returns Yes in 3 out of 5 outputs.

## 6 Evaluation

Our evaluation is designed as an initial demonstration of our novel pedagogical approach, focusing on its feasibility, practical implementation, and impact within a real-world classroom setting. The study was conducted in an undergraduate course on Computer Organization and Assembly Language at CUNY Queens College. To establish a clear baseline, the historical "before" cohorts engaged with a stable curriculum of traditional problem sets and extensive programming projects. Our sole intervention in the "after" cohort was the addition of four assignments using the Socrates system, completed prior to the regular module

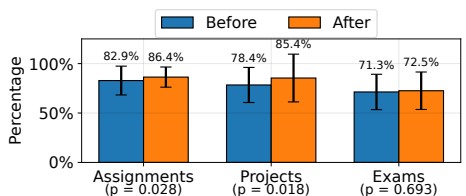

Figure 2: Comparisons of student performance before and after using our approach. Error bars represent the standard deviation.

assignments. Given the practical constraints of a live course, our evaluation employs a quasi-experimental design, comparing student outcomes against these historical cohorts on the otherwise identical coursework rather than using a concurrent, randomized control group. While this limits the generalizability of our findings, it allows us to assess the framework's potential and highlight key considerations for its deployment, such as system cost and LLM performance.

### 6.1 Student Outcomes

We designed questions[1] for the three modules for the class, and deployed them in four assignments. To evaluate the impact, we compared student performance in assignments, projects, and exams, where the number of students (N) ranged from approximately 49 to 80,

---

[1]The case study and detailed description of such questions can be found in Appendix A and B.

against historical cohorts (N = 133 - 237), ensuring only unchanged coursework was used for a fair comparison.

The results are illustrated in Fig. 2. Applying our approach led to a statistically significant improvement in student performance on both Assignments (p = 0.028) and Projects (p = 0.018). These p-values are below the standard 0.05 threshold for significance, indicating the observed improvements are unlikely to be due to random chance. This increase is particularly notable for projects, where students must write code in a hardware description language and an assembly language. Conversely, the observed score increases for Exams (p = 0.693) were not statistically significant. This is likely because the intervention "dosage" (four assignments) had a more localized effect that did not register as strongly on broader measures. Exams assess a wider variety of skills, diluting the specific impact of the intervention. The significant gains on assignments and projects, however, provide strong evidence that the "learning by teaching" paradigm can effectively improve students' mastery of core course competencies.

## 6.2 LLM Performance

To measure the performance of the LLM when they are asked to solve the questions above, we keep logs recording the performance metrics in the playground and the grader. We used gpt-3.5-turbo, and applied gpt-4o and gemini-1.0-pro in the grader on students' submissions of the four assignments. Through such results, we demonstrate the performance of LLMs as they perform in the playground and the grader as follows.

We compare the costs of different LLMs in the playground and the grader in Fig. 3a. We can see that except gpt-4o which is expected to be more intelligent and much more expensive, all other models incur a very low amount of costs which can be easily covered by an instructor. Even gpt-4o consumes only $169.90.

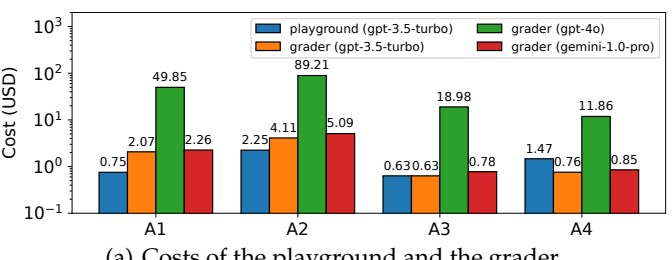

(a) Costs of the playground and the grader.

Fig. 3b demonstrates the correctness of the grader. We compare the results from the grader with those from manual grading. We combine the percentages of true positives (actual correct answers graded to be correct) and true negatives (actual incorrect answers graded to be incorrect), and compare the correctness of gpt-3.5-turbo, gpt-4o, and gemini-1.0-pro. In almost all assignments (except A3), gpt-4o demonstrates the

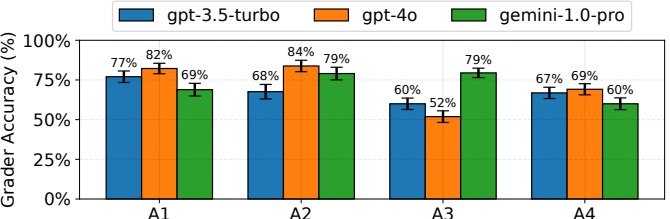

(b) Correctness of the grader, where error bars represent the standard deviation.

Figure 3: Comparisons of LLMs' performance in the playground and the grader.

highest correctness. However, as a cost-benefit observation from our data, the modest performance gain comes at a substantially higher cost, which may make it an impractical choice for budget-constrained educational applications. Interestingly, for assignment A3, which focused on applying transformations in boolean algebra, gemini-1.0-pro achieved much higher correctness than both gpt-3.5-turbo and gpt-4o. This is a tentative observation specific to our test cases, but it suggests a potential strength for gemini-1.0-pro on this particular type of logical reasoning problem.

# 7 Discussion, Limitations, and Future Work

Several important considerations frame this work and point toward future research directions. A primary concern is the risk that students might "game" the system rather than internalizing the content. This could range from simple trial-and-error prompting to a more sophisticated strategy of instructing the LLM to map the novel problem onto a known equivalent. Our methodology includes several layers of mitigation. First, our assessment focuses on the quality and clarity of the student's explanation within the novel context, not merely on achieving a correct final output. A student who simply remaps the problem would likely receive a lower score for an insufficient explanation. Furthermore, the question designs themselves, requiring a coherent mental model to solve, make superficial prompting less effective. Most importantly, our evaluation suggests this engagement fosters genuine learning; the statistically significant gains indicate that the understanding students developed was both transferable and robust, not just a task-specific skill. However, designing questions that are robust against all forms of sophisticated gaming remains an open challenge.

A key methodological choice in this paper was using re-encoded problems to create an information dependency. The pedagogical rationale for this strategy is to foster a deeper engagement with first principles, as students must deconstruct and explicitly define concepts rather than relying on rote memorization. While our results suggest this approach successfully promotes a transferable understanding, we acknowledge that this study does not isolate the specific cognitive effects of re-encoding versus other factors. A focused empirical study directly comparing learning outcomes from re-encoded versus traditional problem formats is a valuable next step. This could lead to exploring more advanced techniques, such as those from machine unlearning, to deliberately suppress an LLM's prior knowledge. Our current prompt-based methodology was intentionally chosen for its practicality and low cost, but the ability to selectively erase knowledge could significantly broaden the applicability of this paradigm, allowing instructors to use state-of-the-art models while still creating the authentic *tabula rasa* effect essential for our approach.

A natural question also arises as to whether our question design strategies can survive the advent of more advanced models, which may solve problems previously considered "LLM-hard." Our framework is designed with this longevity in mind. Instructors can use Socrates to select less-capable models, preserving the pedagogical knowledge gap, and we assume a controlled environment to mitigate the use of external LLMs. The system also allows instructors to append a hidden prompt to the student's submission, which can introduce specific constraints or personas that an external, powerful LLM would not anticipate. This ensures the methodology's longevity, as it depends on the student's direct instruction rather than the LLM's raw reasoning capability.

Finally, we acknowledge the limitations of this study. It lacks a concurrent control group and long-term retention analysis. Future work should conduct larger-scale, controlled experiments to generalize these findings. Exploring the adaptation of this framework to other disciplines also presents a promising research direction. Our design strategies can be adapted accordingly: Strategy 1 is well-suited for novel thought experiments or logic puzzles, while Strategy 2 can also be applied to constructing proofs in STEM or even outlining critical analyses in the humanities. The primary challenge for cross-disciplinary scaling lies in the creation of effective, domain-specific question sets by educators, representing an exciting avenue for future research.

# 8 Conclusions

In this paper, we presented a novel educational methodology that reverses the traditional student-tutor roles by having students instruct an LLM. Our framework, implemented in the Socrates system, uses specifically designed "LLM-hard" questions to foster a more engaging and active learning environment. Our evaluation shows that this approach not only mitigates the risk of student over-reliance by compelling them to produce solutions, but also leads to performance gains on core coursework, demonstrating its pedagogical value and practical viability.

## Ethics Statement

This research involved the evaluation of a novel pedagogical approach using LLMs within an undergraduate computer science course on Computer Organization and Assembly Language. The primary goal was to assess the potential impact of this approach on student learning outcomes.

To evaluate the effectiveness of the methodology, we utilized routinely collected educational data. Specifically, we analyzed aggregated, anonymous student performance metrics, including average grades on coursework (assignments, projects, and exams), comparing the cohort exposed to the new methodology with cohorts from previous semesters where the methodology was not used. Additionally, we examined aggregated, anonymous data from standard end-of-semester course evaluations, focusing on average scores related to the learning experience.

Prior to analyzing this data for research purposes, clarification was sought from the institutional body responsible for research ethics oversight at CUNY Queens College. Following institutional guidance, it was determined by the relevant institutional official that formal Institutional Review Board (IRB) approval was not necessary for this study. This determination was based on the fact that the research relied exclusively on pre-existing, anonymous, and aggregated data typically used for course assessment and improvement, and did not involve interaction with human subjects specifically for research data collection, nor did it involve access to or reporting of any personally identifiable information.

The implementation of the LLM-based teaching activities was integrated into the regular curriculum and assignments for the course. Student participation was part of their standard educational engagement. No individual student data is reported in this paper. All presented results reflect class-level averages and trends, ensuring student anonymity and confidentiality.

Our analysis was conducted with the aim of contributing to the improvement of computer science education practices, while respecting student privacy and adhering to institutional guidelines regarding the use of educational data.

## Reproducibility Statement

This statement provides information regarding the reproducibility of the research presented in this paper, covering the data, code, models, and methodology used. Our aim is to facilitate understanding and potential replication of our work where feasible, while respecting ethical and privacy constraints. The source code and all public data artifacts we share are available at: https://github.com/junli-cuny/Socrates.

**Data.** The evaluation relies on two main types of data. Firstly, student performance data was used, consisting of aggregated, anonymous student grades on coursework (assignments, projects, and exams), as detailed in Section 6. Due to student privacy regulations and ethical considerations outlined in the Ethics Statement, the raw, disaggregated student data cannot be publicly shared. The paper, however, describes the nature of the aggregated data used and the comparative analysis performed (Section 6). Secondly, LLM interaction data, specifically logs of interactions within the Socrates system (student prompts, LLM responses during assignments and grading), were collected. Raw logs containing potentially identifiable student inputs cannot be shared, but the structure of prompts, the interaction flow, and the types of responses generated are described in Sections 4 are publicly shared. Representative examples of designed questions are provided in Appendix B. Regarding assignment/question design, the strategies are detailed in Section 4.1, and the JSON format used within Socrates is described in Section 5. We make anonymized examples of the assignment JSON files available to illustrate the structure and types of questions used.

**Code.** The core software component developed for this research is the Socrates system, comprising the Playground and Grader modules (Section 5). This system is primarily implemented in Python, utilizing Jupyter notebooks, Voilà for the web UI, and standard libraries for interacting with LLM APIs. We release the source code for the Socrates framework

(excluding sensitive API keys and specific course passcodes) under an open-source license. Setting up and running the system will require obtaining API keys from the respective LLM providers (*e.g.*, OpenAI and Google) and installing necessary Python dependencies, details of which will be provided in the repository. Scripts used for generating the figures in Section 6 from aggregated data will also be included in the repository.

**Models and Environment.** The specific LLMs used in our evaluation were OpenAI's `gpt-3.5-turbo`, `gpt-4o`, and Google's `gemini-1.0-pro`. These were accessed via their respective APIs for comparative analysis. Reproducing the exact LLM outputs may be challenging due to the proprietary nature of these models, potential updates made by the providers over time, and inherent stochasticity in model responses. The performance reported reflects the model versions available at the time of the experiments. Accessing these models requires accounts with the providers and may incur costs based on API usage. No specialized hardware is required beyond standard computing resources capable of running Python and accessing web APIs.

**Methodology.** The pedagogical approach, question design strategies, prompt engineering techniques, system architecture, and evaluation methods are described in detail within the main body of the paper. We believe these descriptions provide sufficient detail for conceptual replication of the educational intervention and evaluation framework.

**Limitations.** Full reproducibility is limited by the inability to share raw student data, the dependency on third-party, potentially evolving, commercial LLM APIs, and the inherent costs associated with LLM API calls. Replication in different institutional or course contexts may yield quantitatively different results, though we expect the qualitative findings regarding the pedagogical approach to be informative.

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

## A   Case Study

This appendix provides a detailed description of how the question design strategies presented in Section 4.1 were applied to an actual undergraduate course on Computer Organization and Assembly Languages. This course covers the principles of computer design and implementation across four modules: introduction, data representation, digital logic, and assembly languages. The following sections describe the specific application of our design strategies to the latter three modules. For the complete text of the example questions discussed here, please refer to Appendix B.

**Data representation.** This module covers the fundamental concepts of number systems, including integer and floating-point representations on various bases. It typically introduces

number systems that are essential in computing and digital electronics. For this module, we design questions by leveraging the strategy of creating non-existing scenarios. Here, we introduced new symbols and bases for number representations (such as replacing digits with symbols or altering the base from binary to a non-standard base like three or four). Even if LLMs know arbitrary bases, they only recognize conventional symbols. Thus, human students must provide explicit definitions for LLMs to solve these problems. Following this idea, we design questions with examples of numbers in an arbitrary number system and those in a decimal number system with the same values, and ask human students to teach LLM to convert one to the other. Meanwhile, these questions compel students to apply their understanding of standard number systems in unfamiliar contexts. It tests students' flexibility in applying basic principles to decode and work with completely new systems, mirroring real-world scenarios where theoretical knowledge must be adapted to novel situations.

**Digital logic.** This module delves into digital logic, exploring concepts such as logic gates, Boolean algebra, and circuit design. These are foundational for understanding how computers process information at the most basic level. Questions of this module emphasize complex mathematical reasoning. For example, we ask human students to instruct an LLM to apply De Morgan's laws in intricate ways or transform logical expressions into sum-of-minterms or product-of-sums forms, instead of giving solutions directly. This approach encourages deep engagement with logical constructs, requiring students to perform sequential transformations akin to debugging or optimizing code in software development. By requiring detailed step-by-step manipulations of logical formulas, these questions prepare students for real-world engineering tasks where precise logic manipulation is crucial for designing efficient digital circuits.

Furthermore, we ask students to describe the design of digital circuits using natural language instead of a hardware description language. This gives them a chance to solve the problem by focusing on the design instead of the language details, significantly flattening the learning curve. By focusing on natural language descriptions, students can concentrate on the underlying concepts of digital logic without the initial barrier of learning a new syntactical language. This method not only makes the material more accessible but also enhances inclusiveness, allowing students from diverse academic backgrounds to participate and succeed. Furthermore, it enables them to grasp the essentials of digital logic and circuit design more quickly, fostering a more intuitive understanding of how digital systems work, which can later be translated into more technical languages as their skills and confidence grow. This pedagogical strategy helps us ensure that all students gain a solid foundation in digital logic, regardless of their prior experience with specific programming or hardware description languages. By removing the need to simultaneously master a complex hardware description language, our approach aligns with Cognitive Load Theory (Sweller, 2011). It reduces the extraneous cognitive load (learning syntax), allowing students to focus their mental resources on the intrinsic challenge of understanding digital logic concepts, regardless of their prior programming experience.

**Assembly language.** This module introduces assembly language, a low-level programming language crucial for understanding how software interacts with hardware. It is commonly used in systems programming, embedded systems, and performance-critical applications. We utilize the strategy of creating non-existing scenarios by having students define a new, hypothetical assembly language. Students design instructions for arithmetic, memory, and control flow using natural language. This approach, far from being unconventional, is a standard pedagogical practice in computer architecture education designed to force students to move beyond merely using an instruction set to fundamentally understanding why it is designed a certain way (Patt & Patel, 2003). By requiring students to think like system architects and articulate the precise semantics of each new instruction, this method fosters a deeper and more robust comprehension of the relationship between software and hardware.

## B Example Question Designs

This appendix provides the complete text of example questions designed for the course on Computer Organization and Assembly Languages, illustrating the application of the strategies discussed in Section 4.1 and Appendix A. The questions are categorized by the course module they belong to.

### B.1 Data Representation

These questions[2] primarily leverage Strategy 1 (Creating Non-Existing Scenarios) by introducing novel number systems with unfamiliar symbols, bases, or encoding rules. This requires students to deduce the underlying principles and explicitly define them for the LLM, preventing the LLM from solving the problem using only its pre-existing knowledge of standard systems.

**Question 1: Novel Base-3 System with Symbols**

**Question Text:** We introduce a novel number system where every distinct symbol denotes a unique digit. Let's observe the provided examples of numbers in this system and their respective values in decimal notation:

- _ = 0
- A = 1
- A_ = 3
- AB = 5

Given these examples, deduce the definitions and guiding principles of this number system.

**Rationale:** This demonstration question introduces a base-3 system using symbols "_", "A", and "B" instead of "0", "1", "2". An LLM cannot interpret this system without explicit rules defining the base and symbol mapping, which the student must provide based on the examples. It forces students to apply base conversion principles in an unfamiliar context.

**Question 2: Modified Hexadecimal Representation**

**Question Text:** Below we define a new number system. Various examples are furnished to demonstrate conversions between this system, binary, and decimal representations. Here are the provided examples:

- Binary to new system
  - 0011 = D
  - 1001 = J
  - 10011000 = JIa
  - 01000001 = EB
- New system to binary
  - AA = 00000000
  - CF = 00100101

Based on these examples, your task is to decipher the definitions and rules of this number system.

**Rationale:** This question modifies hexadecimal by mapping digits 0-9 and potentially A-F to a different set of symbols (*e.g.*, A=0, B=1, ..., J=9, then potentially other symbols). The LLM requires the student to explicitly define this non-standard mapping (Strategy 1) to perform conversions based on the provided examples.

**Question 3: Signed Binary with Letter Case**

---

[2]Questions in this module were given to students in two assignments.

**Question Text:** Below we present another new number system. The examples given below demonstrate numbers in this system alongside their corresponding decimal values:

- AABB = 3
- BABB = 11
- aabb = -3
- baba = -10
- AAAA = 0
- aaaa = 0
- ABAB = 5
- abab = -5
- BBBA = 14
- bbbb = -15

From the provided examples, your task is to decipher the definitions and rules of this number system.

**Rationale:** This uses Strategy 1 by replacing "0" and "1" with "A" and "B", and introducing a novel sign encoding based on letter case (*e.g.*, uppercase for positive/zero, and lowercase for negative, possibly sign-magnitude or a similar scheme). The LLM needs the student to define both the digit mapping and the case-based sign convention derived from the examples.

**Question 4: Mixed Base Representation**

**Question Text:** Following examples show numbers in a new base system.

- 1.1 new base = 1.25 decimal
- 10.01 new base = 2.0625 decimal
- 101.001 new base = 5.015625 decimal

Based on these examples, please figure out how a number with the new base can be converted into a decimal.

**Rationale:** This employs Strategy 1 by defining a non-standard mixed-base system (likely base-2 integer part, base-4 fractional part) without explicitly stating the bases. The student must infer these rules from the examples and communicate them clearly to the LLM for conversion.

**Question 5: Custom Floating-Point Format**

**Question Text:** Following examples show a new method to represent the floating-point number with 8 symbols. Based on these examples, please figure out the underlying rules of the new method.

- BAAA0000
  - $+1 \times 2^0 = 1$
- Baai0100
  - $+1.25 \times 2^{-8} = 0.0048828125$
- bABJ0000
  - $-1 \times 2^{19} = -524288$
- baac1000
  - $-1.5 \times 2^{-2} = -0.375$

Hint: The new representation is not biased compared with the traditional one.

**Rationale:** This question uses Strategy 1 to define a completely custom 8-symbol floating-point format. It likely uses letters for sign (B/b), a multi-symbol non-biased exponent (with A-J mapping and case for sign), and an encoded/implicit mantissa. The student must reverse-engineer the complete format (sign representation, exponent base/encoding, mantissa representation) from the examples and explain it step-by-step to the LLM.

## B.2  Digital Logic

These questions emphasize Strategy 2 (Involving mathematical reasoning) and sometimes Strategy 1 by requiring natural language descriptions of circuits. Students must guide the LLM through multi-step logical manipulations or circuit design logic.

**Question 1: De Morgan's Law Application**

**Question Text:** Using De Morgan's law, demonstrate why a xnor b = not(a xor b).

**Rationale:** This demonstration question requires guiding the LLM through a step-by-step proof using Boolean algebra rules (Strategy 2), specifically De Morgan's law. The student provides the reasoning path, defining the intermediate steps for the logical derivation.

**Question 2: Sum-of-Minterms Conversion**

**Question Text:** Convert y = a(b + bc') to the sum-of-minterms form.

**Rationale:** Requires the student to instruct the LLM on the algebraic steps needed for conversion (Strategy 2), such as applying distributive laws, introducing missing variables (*e.g.*, by ANDing with $(c + c')$ which equals 1), and removing duplicate terms. The LLM typically needs explicit guidance for these multi-step transformations to reach the canonical form.

**Question 3: Product-of-Maxterms Conversion**

**Question Text:** A sum term is an ORing of (one or more) variables, like (a + b' + c'). A sum term is sometimes called just a term. An expression in product-of-sums (POS) form consists solely of an ANDing of sum terms, like (a + b' + c)(a + b).

A maxterm is a sum term that has all of the function variables exactly once in either true or complemented form. Product-of-maxterms form is a canonical form of a Boolean equation where the right-side expression is a product-of-sum with each sum consisting only of maxterms.

Convert a'b+ac to its POM form.

**Rationale:** Similar to Question 2 but for the dual canonical form (POS/POM). It requires applying different algebraic manipulations (Strategy 2), often involving principles like duality, starting from the complement, or converting from a truth table, which the student must explain step-by-step to the LLM.

**Question 4: Generalizing a Boolean Function from a Truth Table**

**Question Text:** The truth table below describes a boolean function $y(y_1, y_2, y_3)$ with 3 input boolean variables.

| $y_1$ | $y_2$ | $y_3$ | $y$ |
|---|---|---|---|
| 0 | 0 | 0 | 0 |
| 0 | 0 | 1 | 0 |
| 0 | 1 | 0 | 0 |
| 0 | 1 | 1 | 1 |
| 1 | 0 | 0 | 0 |
| 1 | 0 | 1 | 1 |
| 1 | 1 | 0 | 1 |
| 1 | 1 | 1 | 1 |

Table 1: Truth table for function $y(y_1, y_2, y_3)$.

Describe a function z with the same functionality for 5 input boolean variables, i.e., $z(z_1, z_2, z_3, z_4, z_5)$.

**Rationale:** This involves recognizing the underlying function implemented by the truth table (a 3-input majority function) and then generalizing that logic to a different number of inputs (Strategy 2). The student needs to identify and explain the majority logic concept to the LLM so it can correctly describe the 5-variable implementation.

### Question 5: Circuit Design using Natural Language

**Question Text:** Define $(c, s) = HA(x, y)$ as a half-adder, such that $x$ and $y$ are two boolean input variables, and $c$ and $s$ are two boolean output variables where $s$ is the sum and $c$ is the carry. Typically, the half-adder can be implemented as $s = x \oplus y$ and $c = x \wedge y$.

Describe how we can use the half-adder(s) to implement a 2-bit incrementer $(c, y_1, y_2) = INC(x_1, x_2)$ such that $(y_1 y_2)_2 = (x_1 x_2)_2 + 1$ and $c$ is the carry.

**Rationale:** This combines Strategy 2 (reasoning about how components connect to achieve functionality) and Strategy 1 (using natural language to describe the circuit structure instead of HDL). The student must break down the incrementer logic (*e.g.*, how to add "1" to a 2-bit number $x_1 x_2$) and explain step-by-step how to realize this logic by interconnecting half-adder instances described functionally.

## B.3 Assembly Language

These questions heavily rely on Strategy 1 (Creating Non-Existing Scenarios) by asking students to define and use a new, hypothetical assembly language, often described in natural language. In this section, instructions defined in previous questions may be used in later questions.

### Question 1: Defining Basic Instructions (Init, Load, Add)

**Question Text:** In assembly language, we use addi, mul, ld, and sw and else to finish simple calculations. In this question, create a new assembly language and an addition instruction that initializes the addresses 5004 and 5012 to $t0 and $t1. Load the value to $t2 and $t3 respectively. Then, add up the value and put the sum in $t4.

| Register file | |
|---|---|
| $zero | 0 |
| $t0 | 5004 |
| $t1 | 5016 |
| $t2 | 16 |
| $t3 | 96 |
| $t4 | 16 |
| $t5 | |
| $t6 | |

Table 2: Register File State for Question 1.

| Address | Data memory DM |
|---|---|
| 5000 | |
| 5004 | 16 |
| 5008 | |
| 5012 | 32 |
| 5016 | 96 |

Table 3: Data Memory State for Question 1.

Note that this is a demo question. Students will be given instructions to complete this question.

**Rationale:** This demo question introduces the core task: defining hypothetical instructions (like "init", "load", "add") using natural language descriptions of their behavior (Strategy 1). The student teaches the LLM the precise semantics of this new language (how instructions affect registers and memory based on the tables), which the LLM then uses to generate code for the specified task.

### Question 2: Defining Division and Store Instructions

**Question Text:** Design instructions for division and store, and develop an corresponding assembly program to complete tasks as follows. Instructions defined in all the above questions can also be used.

- Save memory addresses 5004 and 5016 to $t0 and $t1, respectively.
- Load integers from the memory addresses above to $t2 and $t3 respectively.
- Divide the above two integers ($t3/$t2) and save the value in $t4.
- Store the result in the memory address 5008.

**Rationale:** Builds on Question 1, requiring students to define more complex instructions ("divide", "store") using natural language (Strategy 1), specifying their exact effects on registers and memory. Students must then write a program using both new and previously defined instructions, demonstrating understanding of the language they created.

**Question 3: Defining a Conditional Instruction**

**Question Text:** Design a new instruction that could check the value of two registers and complete the following requirements. You can continue using the assembly language created in the last question. Be sure to enter the definition and steps in the required area.

- Add up the integers saved in registers $t2 and $t4 and compare the sum with value in register $t3.
- If the sum is less than the value, find the difference and save the difference in the register $t6. Otherwise, leave $t6 unchanged.

**Rationale:** Focuses on defining conditional logic (Strategy 1). The student must specify the semantics of a new comparison and potentially conditional execution or branching instruction in natural language for the LLM to understand and apply correctly to implement the if-then logic.

**Question 4: Defining a Loop Structure**

**Question Text:** A for loop executes a section of codes repeatedly until it meets a certain condition. In this question, using the logic of a loop, design a new instruction and complete the following requirements.

- While i is less than 5, double variable x.
- Variable i, x are in $t0 and $t1 respectively.
- $t0 = 0, $t1 = 2, $t2 = 5.

**Rationale:** Requires defining instructions necessary for implementing loops (*e.g.*, a conditional jump, decrement/increment, or a specialized loop instruction) using natural language (Strategy 1). The student must explain how these instructions work together to create the iterative behavior needed for the LLM to generate the correct loop code.

**Question 5: Array Initialization using Defined Instructions**

**Question Text:** Define an instruction for multiplication and develop a program using all instructions defined so far to create an array with 3 32-bit integers. The value of each elements should be the square of the index (starting from 0). The base address of this array is 5000.

**Rationale:** Combines previously defined concepts (loops, arithmetic, memory access via "store") and potentially requires a new "multiply" instruction definition (Strategy 1). The student must structure the program logic (looping through indices, calculating the square, storing to the correct memory address) using the custom language they have defined for the LLM.

**Question 6: Array Processing (Min/Max) using Defined Instructions**

**Question Text:** Design a program that finds the minimum and maximum value of an array with 32-bit integers using the instructions created above. The base address of the array is 5000, and the size of the array is 4. Save the result in the register $t0.

**Rationale:** This acts as a capstone, requiring students to apply the full set of previously defined instructions to implement a more complex algorithm (finding min/max in an array)

purely within the student-defined assembly language. No new definitions are needed, focusing on demonstrating mastery of applying the created language constructs (loops, comparisons, loads/stores) to solve the problem.

