# OpenReview forum: "Learning by Teaching: Engaging Students as Instructors of Large Language Models in Computer Science Education"
_colmweb.org/COLM/2025/Conference — COLM 2025_

### Official Review · Reviewer_4NdV · 2025-05-08

**Rating:** 6
**Confidence:** 4
**Ethics Flag:** 1

**Summary:**

This paper proposes an educational use of large language models (LLMs) with limited capabilities by reversing the typical communication flow: instead of students asking questions and the LLM responding, students are tasked with teaching the LLM to solve problems. This is achieved through prompt engineering rather than dialogue, requiring students to understand and explain problems clearly in order to elicit correct outputs from the model. To prevent the LLM from drawing on prior training knowledge, standard problems in introductory computer science courses (e.g., number representation, digital logic, assembly language) are modified through re-encoding. The authors also use a second LLM to evaluate whether the "student" LLM arrives at the correct solution, and deploy the approach in a real undergraduate course using their Socrates platform, reporting slight improvements in learning outcomes compared to previous semesters.

**Questions To Authors:**

* Some references are missing publication years (e.g., Liu or Markel or Jeon & Lee in section 3).
* The paper remains too conceptual and delays concrete examples until the appendix. Bringing selected examples into the main text (e.g., into the introduction or Section 4) would help readers better understand the methodology earlier.
* Future work could explore more advanced techniques for selectively suppressing LLM prior knowledge. A core challenge not addressed is how to deliberately erase or mask an LLM's prior knowledge to enable broader use of stronger models while maintaining the tabula rasa effect. This connects with the machine unlearning approaches and could be explored in future papers.

**Reasons To Accept:**

* The paper addresses growing concerns about student overreliance on LLMs by designing an approach that promotes active learning instead of passive one.
* It implements the educational principle of "learning by teaching" which is difficult to scale in traditional human-based settings but becomes feasible using "dumb" LLMs.
* The methodology was tested with real students, showing measurable (albeit modest) gains in performance.
* The use of a secondary LLM as a grader enables scalable, low-cost assessment.
* Problem re-encoding is an interesting approach to knowledge cleaning in LLMs.

**Reasons To Reject:**

Not necessarily reasons to reject, but some of the following points could be addressed:

* The pedagogical effects of re-encoding standard problems to cancel the LLM's prior knowledge are not fully explored. Such transformations could either enhance or hinder human student understanding, and further empirical study is needed.
* There is a risk that students might "hack" the system by prompting the LLM in ways that reveal the problem's equivalence to a known one.
* The paper reports student outcome improvements but lacks details on sample size or statistical significance testing, making it hard to assess the robustness of the findings.
* Students may struggle to interact with lower-capability LLMs if they are accustomed to more advanced current models, introducing a learning overhead not addressed in the paper.
* The pre-existing conditions of the course used for comparison are not clearly described, making it difficult to assess the strength of the before/after comparison. Which activities had the students to do the previous year?
* Because LLMs are highly sensitive to prompt phrasing, students may experience frustration when otherwise correct explanations fail to be understood by the model, requiring potentially artificial prompt tweaking.

---

> ### Author Response · Authors · 2025-06-02
>
> On the pedagogical effects of re-encoding problems, our rationale is that novel formats compel deeper engagement with underlying principles, as students must deconstruct, adapt, and explicitly instruct the LLM, preventing rote application. This design also supports a low-cost, easy-to-deploy system, avoiding complex techniques like fine-tuning. Our evaluation shows our overall approach yields statistically significant improvements in regular coursework performance, suggesting the deeper engagement fostered is beneficial. While the study doesn't isolate re-encoding's specific cognitive effects, the positive transfer indicates overall framework effectiveness. We will clarify that a focused empirical study on re-encoding's impact is valuable future work.
>
> Regarding the risk of “hacking” by remapping novel problems to known ones, this is a perceptive comment. Interestingly, correctly identifying and articulating such an equivalence itself requires understanding. Our design aims to make simple remapping less effective; novel scenario complexity should make direct translation difficult. Assessment emphasizes explaining the solution within the new context. If instruction is mainly remapping, it may not suffice. Socrates saves interaction logs, allowing instructors to check. However, the risk is valid. Effective question design must make novel aspects integral. We will discuss this "gaming" risk as a challenge and limitation.
>
> Concerning the lack of detail on sample size or statistical significance for student outcomes, we will revise Section 7.1 to include student numbers for 'before' and 'after' cohorts. We have now conducted statistical tests, and will report means, standard deviations (Fig 2), the test used, and p-values. Our preliminary analysis shows significant improvements for 'Assignments' (p≈0.028) and 'Projects' (p≈0.018), as "Learning by Teaching" LLM assignments foster deep understanding directly applicable to homework/projects. Differences for 'Exams' (p≈0.693) and course 'Evaluations' (p≈0.781) were not statistically significant, as ‘Exams’ are broader and test varied skills, diluting the impact, and ‘Evaluation’ are multifactorial. The intervention "dosage" (four Socrates assignments) might also yield more localized effects. The significant improvements in assignments/projects suggest our approach has enhanced complex task ability. Non-significant exam/evaluation results may mean that the impact was less direct or that these measures are influenced by wider factors.
>
> On students struggling with lower-capability LLMs, our approach intentionally uses LLMs requiring careful prompting for students to be effective instructors. We acknowledge potential challenges (lines 249-251) and provide prompt engineering guidance. Socrates allows instructors to select LLMs, balancing "difficulty" with budget and objectives. If students accustomed to advanced LLMs find instructing a less capable "virtual student" challenging, this can be pedagogically valuable, demonstrating capabilities taken for granted. This "struggle" is central, forcing refinement and deeper thought, countering over-reliance.
>
> Regarding pre-existing course conditions for comparison, the course is "Computer Organization and Assembly Languages." In previous semesters, evaluated components (Assignments, Projects, Exams, Evaluations) were substantially the same. Assignments were traditional problem sets. Projects were extensive programming. Exams had a consistent format/coverage. Evaluations were standard student feedback forms. The main intervention was adding four Socrates assignments before regular module assignments. Regular coursework used for comparison remained unchanged in content/difficulty.
>
> Concerning student frustration with prompt phrasing, our paper acknowledges this (lines 250-252) and details guidance (Sec 5), including encouraging student CoT and using LLM self-consistency. CoT encourages logical structure. Self-consistency (multiple LLM attempts, success threshold) helps interpret imperfectly phrased but conceptually sound answers. While the line between useful refinement and "artificial tweaking" is subtle, less capable LLMs necessitate more fundamental explanations, making superficial tweaks less likely to succeed. The ideal is that tweaking drives precise articulation. System logs allow instructor review.
>
> For your questions: 1. Missing publication years (e.g., Liu, Markel, Jeon & Lee in Sec 3) will be added. 2. To make the paper less conceptual earlier, we will bring concise examples of our question design strategies (from Appendix A) into the main text (e.g., Sec 4.1, 4.2). This will balance illustration with keeping the appendix for extensive details. 3. Exploring advanced techniques like machine unlearning to selectively suppress LLM knowledge is an insightful suggestion for future work. Our current prompt-based approach prioritizes low cost and ease of implementation.

---

> > ### Comment · Reviewer_4NdV · 2025-06-09
> >
> > I acknowledge and appreciate the authors’ response. As my scores are aligned with those of the other reviewers, I maintain my original score.

---

> > > ### Author Response · Authors · 2025-06-10
> > >
> > > Thank you! We appreciate your thoughtful review and constructive feedback throughout this process.

---

### Official Review · Reviewer_q5Y1 · 2025-05-11

**Rating:** 6
**Confidence:** 3
**Ethics Flag:** 1

**Summary:**

This paper presents a novel approach to using large language models (LLMs) in computer science education by reversing the traditional student-LLM roles. Instead of treating the LLM as a tutor, students are asked to teach the LLM to solve problems, requiring them to deeply engage with the material. The authors implement this methodology in Socrates, a system that facilitates prompt-based instruction, self-consistency evaluation, and CoT-based student prompting. They deploy this system in a real course and report improved student outcomes and low cost. The idea is good, well-motivated, and practically implemented. The writing is generally clear, though at times a bit redundant. The evaluation, while not large-scale, offers reasonable evidence of learning gains.

**Questions To Authors:**

1. Have you considered how this method would scale across institutions or disciplines outside CS? Some discussion might needed in the paper.

2. Is there a risk that students simply “game” the LLM rather than internalizing the content? How do you measure true understanding? For example, students might iteratively trial-and-error their prompts or rely on memorized CoT structures without fully understanding the underlying concepts.

**Reasons To Accept:**

**1. Novel Pedagogical Model:** Reversing the roles between student and LLM is an innovative framing that encourages deeper learning and prompt literacy.

**2. Low-cost & Scalable:** The method relies on affordable LLMs (e.g., GPT-3.5) and supports structured prompting (CoT, few-shot, self-consistency) to make weaker models viable.

**3. Rich Methodology:** The paper provides detailed strategies for designing questions that cannot be easily solved by the LLM alone, ensuring students must contribute meaningful content.

**4. Good Writing:** Overall, this paper is well-written and I could easy follow the logic of this paper.

**Reasons To Reject:**

**1. Limited Evaluation Scale:** The study involves a single course and lacks a more rigorous experimental setup (e.g., control groups, long-term retention).

**2. Technique Contribution is limited:** Technique contribution is limited. Overall, the work is mostly prompt tuning.

---

> ### Author Response · Authors · 2025-06-02
>
> We thank Reviewer q5Y1 for their positive assessment of our work, recognizing its novelty, practical implementation, and clarity. We will address the points raised.
>
> (#1, Limited Evaluation Scale) We understand this concern. Our paper introduces a novel pedagogical approach ("Learning by Teaching" an LLM) and the Socrates system. The single-course case study (undergraduate Computer Organization & Assembly Language) serves as an initial demonstration of feasibility, positive impact in this context, and practical aspects like ease of use and low cost (Sec 6, 7.2), crucial for wider adoption. We acknowledge this study lacks a concurrent randomized control group or long-term retention analysis. Such broader evaluations are important next steps. We will articulate these as limitations and future research directions, framing our current evaluation as a foundational demonstration of this promising, implementable approach.
>
> (#2, Technique Contribution is Limited) We appreciate this perspective. Our work emphasizes applying and structuring LLM interactions for education, rather than new model architectures, aligning with COLM’s scope. Our contributions include:
> 1. Novel Pedagogy: Reversing student-LLM roles fundamentally alters interaction for deeper learning.
> 2. LLM-Aware Question Design: Strategies 1 (non-existing scenarios) & 2 (mathematical reasoning) create tasks initially challenging for LLMs, requiring student scaffolding. This deliberate design, understanding LLM capabilities, is a key technical element.
> 3. Socrates System: Designed for easy deployment by instructors, with features for controlled interaction and LLM-based grading. Its architecture is tailored to our educational goals.
> 4. Pedagogical Prompt Engineering: We guide students to teach an LLM, distinct from merely using an LLM for answers. The student's instructional "prompt" is the core learning activity.
> 5. Low-Cost Accessibility: Our method avoids complex, costly techniques like fine-tuning. The novelty lies in structuring interactions within an educational framework where the student is the instructor. The technical challenge is designing questions, the system, and interaction flow for improved, cost-effective learning. We will clarify these distinct contributions, emphasizing pedagogical innovation and practical design.
>
> (Question 1, Scalability Across Disciplines) This is an excellent question. While our study is in CS, the core principle of solidifying understanding by explaining to a virtual protégé (LLM) has broader applicability. In math, students could teach problem-solving procedures; in history, explain event causality; in language learning, define grammar. Adapting question design (Strategy 1 for novel counterfactuals/puzzles; Strategy 2 for guided reasoning in STEM or critical analysis) is key. Socrates' flexible JSON input could facilitate this. The main challenge is educators developing effective, discipline-specific question sets. We will add a discussion on this potential for scalability and cross-disciplinary use, noting needs for further research and adaptation.
>
> (Question 2, Risk of "Gaming" & Measuring True Understanding) This is an important concern. We acknowledge the risk of students superficially "gaming" the LLM. Our mitigations include:
> 1. Question Design (Sec 4.1): Strategy 1 (novel scenarios, e.g., new number systems) requires genuine understanding to define rules for the LLM, making minor prompt tweaks insufficient. Strategy 2 (guided multi-step reasoning) pushes beyond superficial engagement.
> 2. Assessment Focus: The student's "answer" is their explanation to the LLM. Assessment focuses on this instruction's quality and clarity. Iterative refinement, if reflective, can be learning.
> 3. Interaction Logs: Socrates saves interactions (lines 300-302), allowing instructor review of the student's teaching process, not just the final prompt.
> 4. Question Complexity: Designed to be non-trivial for LLMs without detailed instruction, making simple gaming less effective.
> 5. Broader Curriculum Assessment: Socrates assignments are part of a course with traditional exams/projects (Sec 7.1). Improved performance on these (Fig 2) suggests transferable, not superficial, learning.
> However, no system fully eliminates superficial engagement. Measuring "true understanding" is challenging. While our method encourages deep engagement via explanation, future research could explore more explicit measures (analyzing prompt evolution, metacognitive prompts, rigorous follow-up tasks). We will discuss this risk, our mitigations, and future research avenues.

---

> > ### Comment · Reviewer_q5Y1 · 2025-06-07
> > **Thanks for the response**
> >
> > Thanks for the response. Most of my concerns are addressed. I will keep my positive score.

---

> > > ### Author Response · Authors · 2025-06-10
> > >
> > > Thank you! We appreciate you taking the time to read our response and for your constructive feedback on our work.

---

### Official Review · Reviewer_eJrt · 2025-05-12

**Rating:** 5
**Confidence:** 4
**Ethics Flag:** 1

**Summary:**

This paper presents work on role reversal (teacher-student) using LLMs in CS education. The paper shows that their method improves learning outcomes compared to a control group.

**Reasons To Accept:**

I like the idea of reversing the roles of student and teacher, as it may engage students more efficiently.

**Reasons To Reject:**

I found the paper rather terse and difficult to decipher; the paper gives a somewhat high-level description of the system but could benefit from more details. I did not quite understand how the two different types of question generation worked, especially the first one. The paper claims that using scenarios or languages unknown to the model at training time will force the model to not know the answers to the question? I also don't quite follow with how the grader works. Why does the grader send prompts to the LLM instead of a teacher looking at the student submissions? The related work section calls itself comprehensive, which I would disagree with. The paper mentions a potential over-reliance of students on LLMs but does not further address this question in its experimental or results section.

The language should also be proof-read.

---

> ### Author Response · Authors · 2025-06-02
>
> We thank Reviewer eJrt for their feedback and positive view on our core idea. We will revise for clarity and detail.
>
> Question Generation & LLM Knowledge: We will clarify our two question generation strategies. Strategy 1 ("Creating non-existing scenarios") introduces novel elements (e.g., new symbols for numbers, unique assembly instructions) not in the LLM's training data. For example, defining a base-3 system with symbols '@', '#', '$' forces students to explain the rules, as the LLM cannot solve it from prior knowledge alone. This ensures students actively construct and convey understanding. Strategy 2 ("Involving mathematical reasoning") uses tasks requiring multi-step guided deduction (e.g., applying De Morgan's laws step-by-step), which less capable LLMs struggle with unassisted.
>
> To be precise about LLMs "not knowing" answers: while an LLM understands general principles (e.g., base conversion), novel specifics (like a new symbolic base-4 system or a custom assembly instruction) are outside its direct experience. It cannot operate within these new constraints without explicit student definitions. Thus, the LLM is "forced not to know" the specific answer, becoming dependent on the student's instruction, which is key to our pedagogical approach requiring deep student understanding. This nuance will be better articulated.
>
> Grader Functionality: The grader (lines 367-378) assists instructors by partially automating assessment. It takes the student's instructions (their answer) from the playground and sends them to an LLM (potentially with hidden test cases) to generate an output. A second LLM interaction then verifies this output against a predefined correct sample, providing a "Yes/No." This automates comparison, aiding scalability, especially with varied but semantically valid answers. This does not replace the instructor, who designs questions, samples, test cases, and criteria. Instructors would still review ambiguous cases, interaction logs (Socrates saves these), and manually grade selections. The LLM grader is an efficient assistant, with instructor oversight essential. This will be clarified.
>
> Related Works & Pedagogical Grounding: We acknowledge "comprehensive view" (line 140) may be an overstatement. We will revise this phrasing and the entire Related Works section (Sec 3) for more substantive, focused summaries. For each cited paper (e.g., Brown & Garcia; Jeon & Lee, 2023; Bastani et al., 2024), we will focus on its primary research contribution, findings, and specific relevance to our role-reversal theme, moving beyond superficial details. We will articulate how prior art informs our understanding of challenges (over-reliance, academic dishonesty, educator misuse) and opportunities. For instance, discussing Jeon & Lee (2023), we'll briefly exemplify inadvertent misuse. Citing Bastani et al. (2024), we'll include their study's high school context.
>
> To strengthen pedagogical foundations (lines 41-45, 148-149), we will cite research on the "protégé effect" (e.g., Roscoe & Chi, 2007; Chase et al., 2009) and Cognitive Load Theory (Sweller, 1988) for lines 227-229. This will better contextualize our approach and demonstrate thorough engagement with relevant literature. All citations will be checked for completeness.
>
> Roscoe, R. D., & Chi, M. T. H. (2007). Understanding tutor learning: Knowledge-building and knowledge-telling in peer tutors’ explanations and questions. Review of Educational Research, 77(4), 534-572.
>
> Chase, C. C., Chin, D. B., Oppezzo, M. A., & Schwartz, D. L. (2009). Teachable agents and the protégé effect: Increasing the effort towards learning. Journal of Science Education and Technology, 18(4), 334-352.
>
> Sweller, J. (1988). Cognitive load during problem solving: Effects on learning. Cognitive Science, 12(2), 257-285
>
> Over-Reliance: While our introduction notes over-reliance concerns (lines 37-40), Section 7 doesn't empirically measure it. Our "learning by teaching" design inherently mitigates in-system LLM over-reliance for answers, as students must instruct an initially "unknowing" LLM. Their task is to generate explanations, not receive solutions. This demands active engagement. We acknowledge students might use external LLMs, but policing this is beyond our current scope. We will revise to state our design's inherent mitigation and note external use as a separate challenge for future study.
>
> Language Proofreading: We will thoroughly review the entire manuscript for language, grammar, and clarity. This includes refining "unsure" language (line 167), ensuring precise descriptions of CoT/BERT, making related work summaries substantive, wording claims about performance/cost carefully, and clarifying explanations of question strategies, the grader, and pedagogical rationale. Vague statements (e.g., intro on cost/answers) will be linked to later substantiation.

---

> > ### Comment · Reviewer_eJrt · 2025-06-09
> >
> > Thank you for the detailed response. I am certain the revisions will improve the paper substantially.

---

> > > ### Author Response · Authors · 2025-06-10
> > >
> > > Thank you! We appreciate your positive follow-up and are grateful for your constructive guidance throughout the review process. We are committed to implementing the revisions as discussed and are confident they will significantly strengthen the manuscript.

---

### Official Review · Reviewer_VpCm · 2025-05-13

**Rating:** 5
**Confidence:** 4
**Ethics Flag:** 1

**Summary:**

Review of "Learning by Teaching: Engaging Students as Instructors of Large Language Models in Computer Science Education"

This paper investigates how large language models can be used to emulate a "student" learning a course, instead of a "tutor" giving lessons or feedback, in order to improving learning outcomes of students by asking them to "explain" the concepts to the LLMs. The authors develop strategies for question design and also implement a system enabling instructors to use their proposed method without necessarily knowing how to code.

**Reasons To Accept:**

The paper focuses on a relevant idea in the domain of using LLMs for education, and the system developed by the authors has merits to be useful in practice by tutors and educational stakeholders. With that said, I believe there are several aspects in the paper that need to be revised, rethought, or improved, before the paper is accepted at COLM.

**Reasons To Reject:**

My points are roughly in order of importance starting from the highest:

1. There is a clear separation between usage of LLMs in CS (e.g., writing code using Copilot) and their usage for "learning" CS, in other words between using LLMs as a "utility" or as a "tutor" but the paper, mainly the intro, currently mixes them up (e.g., line 31).
2. A major issue in the paper comes from how the questions have been designed. The authors claimed they have designed hard questions that cannot be solved by an LLM normally. However, this would inflate the difficulty level of the questions, possibly harming student learning. Why was not the LLM just prompted to be deliberately "dumb" and simulate a real student? Claims such as the one on line 150 are not necessarily going to stay true in the long term, as LLMs might improve over time.
3. Relatedly, having only difficult questions also biases human performance and prevents it to reflect real-world performance, which makes the analysis unfair when considering real-world environments.
4. "Guiding students to give answers following CoT": this suggests that the human, i.e., student, should do CoT. But I assume there are better proven educational strategies for humans, why should they do CoT?
5. Relatedly, the framing of the paper shifts towards teaching students prompting strategies rather than helping them explain concepts to the LLM for improved learning. This is against the main motivation of the paper and is not relevant to the current study; this major discrepancy should be addressed and elaborated properly.
6. Regarding the system design section: first, it is not clear from where and how the requirements in lines 297 to 302 have been extracted. Second, the paper seems to lack an evaluation of the design aspects of the system in particular (which can be fine but needs to be explicitly said as a limitation or suggestion for future work).
7. Another issue that can be found in the paper lies in the evaluation. It seems that there has been no control group, which limits the claims the authors can make about their findings (at least it should be mentioned as a limitation; the limitations section is too short now). Moreover, the standard deviations in the plots (as well as proper statistical tests) are missing from the evaluation, limiting us to see if the differences were significant.
8. Relatedly, line 407 claims the difference has been "significant" while there is no statistical test involved. This is an important mistake to avoid.
9. I don't find what is new regarding the contributions of the paper in the "We compare the costs of" paragraph in section 7.2. To me, it seems like a side experiment/evaluation which doesn't have much to do with the main story of the paper (so can go to the appendix). Also, I don't find what new information this brings to us, as it can already be found on the pricing page of LLM providers.
10. It is not clear if the work in line 137 actually covers using LLMs as a tool or as a tutor.
11. Lines 148-149 should cover works that claim this is a beneficial thing to also do with humans. Also, it is beneficial to mention at which stage of grading, which grade/level, etc.
12. "generate multiple responses with the same prompt": why this is going to be helpful, if the goal is to teach students to ask better questions?
13. The paper provides the idea of students defining new languages and giving the definition to the LLM. Is that something that happens regularly, or has been shown to improve student learning? Otherwise, why do the authors think it is a good strategy to follow?
14. Relatedly, is "defining new assembly language instructions" a correct way to do this? Isn't this against the principles of teaching a programming language and asking the students to solve the problems using the same language set?
15. The paper mentions JSON as a way to reduce the necessary knowledge of programming by the tutors. However, Socrates could have had a user interface for directly adding the designed questions, removing the need for any programming knowledge altogether. Was there any reason that this has not been explored?
16. There is no source provided for the claim on lines 227-229.
17. Relatedly, he paper touches on an educational topic, so it has the possibility to connect with educational or learning sciences literature on the benefits of explaining a concept to others on one's own learning. However, this is currently not sufficiently covered in the paper. I believe the paragraph on lines 41-45 are not shown to be true first time in this paper, but it lacks citations.
18. I would organize section 2 better by a proper sub-sectioning.
19. I would say that section 2, as well as a lot of parts in the paper, touch on the basics of LLMs, CoT, etc and they are unnecessary for a COLM submission. Similarly the first few lines of section 3, lines 245 or 246, or lines 253 to 256.
20. "much more expensive": this is something that the models can improve upon over time (by new hardware) regarding the scale of the current work, so I don't believe that can be justified as a motivation.
21. "they think they are sufficient, with no correct output from the LLM": the evidence backing this sentence is not clear.
22. "However, it is not high enough to justify its high cost": this claim is not supported properly by the data, i.e., it's too bold of a claim in that context.
23. The sentence in line 446 can't be deduced based on a single data point, unless evidence is provided from the literature.
24. I would reduce the amount of information in section 4.2 as I think it is unnecessary to cover all of them in the body of the paper (might be find for appendix, however).
25. The paper has an "unsure" language in some parts, e.g., "it can make the LLM unable to apply" on line 167.
26. I find it unnatural to group BERT with the other LLMs in the same paragraph (and find it irrelevant to the current study). To me, it seems more like a generic explanation of LLMs, which is not necessary for a COLM submission (see my previous point).
27. It is not related how the risk of academic dishonesty is related to "learning" as it refers to LLM as a tool, not as a tutor.
28. Figure 1 is overly simple; for a COLM paper, it would be expected to have a figure showing details of the NLP pipeline, rather than the overly-simplified system architecture.
29. Is the "prompt" in line 375 a CoT prompt? If not, why?
30. Unclear and vague sentence: "The system incurs a low cost with a satisfying level of answers"
31. I would be more careful with calling CoT a "recent advance" or with calling BERT "revolutionized".
32. It is unclear why using LangChain was an important aspect to cover from the paper by Brown and Garcia: to me it seems like the one-line summaries LLMs would generate given a paper, so I would suggest a deeper dive and providing the actual contribution of each paper instead.
33. How educators may "inadvertently misuse" LLMs should be described (line 135).
34. The title of section 4 has a mismatch with the topic sentence of the first paragraph. Relatedly, the first sentence of this section is redundant when compared to the last sentence of the previous section. A similar redundancy happens on lies 162-163.
35. Some citations have missing information, e.g., on lines 109, 134, and 143.
36. Personal opinion: I would have motivated the contents of section 7 earlier, already from the introduction. However, I will leave this to the authors to decide. Additionally, based on the type of your study, I would suggest providing clear research questions in the introduction, but again it is up to the authors.

I thank the authors for their submission and I hope my review would be useful for improving their paper.

---

> ### Author Response · Authors · 2025-06-02
>
> (#2, 3) Our questions are "hard" for an LLM without student-provided context, not for students. This necessitates students to provide the missing steps to an LLM, a valuable learning step. Despite LLM evolution, Strategy 1 using less capable LLMs (instructor-selectable) maintains pedagogical value. Our evaluation compares performance on other, regular coursework, showing positive transfer from the additional "teach the LLM" tasks, not performance on the novel tasks.
>
> (#4, 5) Student CoT helps the LLM process instructions. Preparing CoT explanations, however, benefits students by requiring problem decomposition and clear articulation, aligning with "learning by teaching." Prompting strategies are not general prompt engineering training but are crucial methods for students to effectively explain concepts and instruct the LLM, directly supporting our core motivation of enhanced student learning.
>
> (#6) System design requirements stem from pedagogical goals like fairness and robust assessment. A specific usability evaluation of Socrates was beyond this initial study's scope and will be future work.
>
> (#7, 8) We acknowledge our comparison uses historical data (a limitation). Statistical analysis now performed: standard deviations for before (Assignments:14.57; Projects:17.76; Exams:17.90; Evaluation:21.00) and after (Assignments:10.20; Projects:24.16; Exams:18.96; Evaluation:22.80) cohorts will be added to Fig 2. Tests show significant improvements for ‘Assignments’ (p=0.028) & ‘Projects’ (p=0.018). For Exams & Evaluations, differences were not statistically significant, as our intervention's impact is more direct on assignments/projects requiring deep understanding similar to our tasks, while exams are broader and evaluations are multifactorial.
>
> (#9) The cost analysis presents contextualized costs for our system and specific assignments, supporting our claim of practical cost-effectiveness beyond generic API pricing, which is crucial for real-world adoption.
>
> (#10, 11) Bastani et al. discuss over-reliance when LLMs are tools, informing our role-reversal. For the sentence mentioned, "beneficial" refers to understanding these risks, which motivates our pedagogical approach. Their study involved ~1000 high school students (9th-11th grade) in math.
>
> (#12, 13, 14) Allowing multiple LLM responses helps distinguish LLM variability from flaws in student instruction, guiding student refinement. Defining novel languages deepens understanding of underlying principles by requiring precise articulation for a new context (constructivism), akin to how computer architects think and design, fostering robust comprehension.
>
> (#15) JSON for question input was chosen for rapid prototyping and flexibility in early research stages, allowing quick iteration. A full GUI, while enhancing usability, was a larger undertaking. Our immediate focus was the pedagogical methodology.
>
> (#16, 17) For lines 227-229, Cognitive Load Theory (Sweller, 1988) supports using natural language to reduce extraneous cognitive load, aiding focus on core concepts. For lines 41-45, we will further cite work on the protégé effect.
>
> (#20, 21) The pedagogical need for weaker LLMs (requiring proper, detailed prompting) is primary for student learning; their current lower cost is a practical alignment. Student frustration is an anticipated HCI challenge motivating our guidance strategies.
>
> (#22, 23) The claim "not high enough to justify its high cost" is a cost-benefit observation from our specific data. We'll rephrase for clarity. Line 446 is not general but a tentative observation from our assignments; this will be clarified as specific to our test cases.
>
> (#27) When LLMs are used as tools for answers, students can bypass crucial learning processes. Our role-reversal makes teaching the task; assessment focuses on the quality and depth of student instruction, thereby mitigating academic dishonesty.
>
> (#28, 29) Our core contribution is pedagogical, not a novel NLP pipeline; Figure 1 reflects this system focus, illustrating interaction flows. The grader prompt for Yes/No verification is for direct classification, not CoT-style step-by-step reasoning.
>
> (#33) Educators might "inadvertently misuse" LLMs from assignments LLMs can easily solve without true student learning, or lack clear usage/assessment policies. Our model aims to mitigate these risks by structuring active student engagement.
>
> Writing/Structure: (#1) Introduction will clarify utility vs. learning use. (#18, #19, #26) Sec 2 will be restructured, LLM/CoT basics condensed, BERT emphasis reduced. (#24) Sec 4.2 will be more concise. (#25) Line 167 will be more confident about strategy effects. (#30) Line 71will clearly foreshadow later results. (#31) Language for CoT/BERT will be revised. (#32) Related Works will be more substantive and analytical. (#34) Sec 4 start will be revised for better flow and no redundancy. (#35) Missing citation years will be added. (#36) Evaluation will be signposted earlier.

---

> > ### Comment · Reviewer_VpCm · 2025-06-05
> >
> > Thanks a lot for your comment on the review.
> >
> > Points 2 and 3 still don't address why having students help LLM in solving a difficult problem would have pedagogical value compared to simpler problems, and whu prompting the LLM to act like a student of level X would not work. Relying on the smaller LLMs not being able to solve difficult problems might be risky, because the types of problems that LLMs get to solve with increasing the scale does not necessarily correspond to problems with increasing difficulty for "humans" to solve.
> >
> > About points 4 and 5: nice explanation in the responses, this should be made way more clear in the paper. This seems to me a research direction of its own (alignment of "strategies of talking to an LLM" with "real-world teaching strategies"), because if these two are not shown to represent the same thing, the "learning by teaching" method, which was originally designed for teaching real students and not LLMs, does not hold. The LLM should be guaranteed to be able to simulate an over-time-learning student properly, in order for this concept to be applicable here.
> >
> > On point 6 make sure you provide the specific citations and resources for the design decisions (rather than only generically explaining the pedagogical goals). Similarly, references are needed for 12-13-14.
> >
> > On points 7 and 8: thanks a lot for the statistical analysis, that is really beneficial to have. Also thanks for explaining certain design decisions properly, e.g., point 15 (remember to add those to the paper as well).

---

> > > ### Author Response · Authors · 2025-06-06
> > >
> > > Thank you for your thoughtful follow-up comments. They have helped us pinpoint key areas for clarification and improvement in our manuscript.
> > >
> > > Regarding your first point on question design, we will clarify that the "hardness" of a problem is intentionally engineered for the LLM, not for the human student. Standard computer science problems often exist in the LLM's training data, allowing it to retrieve answers without the reasoning process that provides pedagogical value. Our strategies create novel problems that are underspecified from the model's perspective, requiring a student to use their course knowledge to articulate the missing context and guide the LLM.
> > >
> > > Crucially, a problem that is hard for an LLM is not necessarily hard for a student. For example, a simple strategy of replacing binary '0' and '1' with 'A' and 'B' can easily confuse an LLM, but a student learning binary representation can readily explain the new convention. To demonstrate that these problems remain manageable for students, we analyzed their performance on our "teach the LLM" assignments. The average student success rate was approximately 70%, with many questions seeing rates well above 80%, confirming that the tasks are appropriately challenging but not overwhelming for learners.
> > >
> > > You also asked why we don't simply prompt the LLM to act as a less capable student. This is a key point, and we will clarify our rationale. Prompting an LLM to consistently and believably simulate a novice can be unreliable, a finding supported by recent works highlighting the inconsistency of LLM-based simulations and their failure to maintain coherent low-proficiency personas (Kumar et al., 2025; Mannekote et al., 2024). Our approach provides a more authentic learning experience, as it requires students to diagnose and remedy a specific, genuine knowledge gap in the model rather than interacting with an arbitrarily simulated persona. Moreover, our system design allows instructors to select an LLM with the appropriate capability for a given question, ensuring the task remains a meaningful teaching exercise.
> > >
> > > Regarding your point on aligning LLM instruction with pedagogy, you have accurately identified the core conceptual assumption of our work. We will address this more directly in our revision. Our paper presents an initial exploration into this alignment, and we will strengthen the argument by explicitly connecting our techniques to established learning theories. For instance, having students provide step-by-step instructions is a direct application of problem decomposition, a fundamental computational thinking skill (Brennan & Resnick, 2012). By articulating these steps, students are not just prompting; they are engaging in a structured problem-solving process that reinforces their own procedural thinking (Lye & Koh, 2014). Similarly, our strategy of having students create novel languages is a direct application of constructivist learning theory (Papert & Harel, 1991; Kafai & Resnick, 1996). By defining the rules of a new system, students build a shareable artifact, a process that forces a more rigorous understanding of underlying concepts than simply using a pre-existing language (Papert & Harel, 1991).
> > >
> > > Finally, we will act on your direct suggestions. We will integrate the new statistical analysis and provide the requested citations to justify our design and pedagogical choices. For the system design (#6), we will ground the requirements for Socrates in established principles of assessment fairness (American Educational Research Association et al., 2014) and academic integrity (e.g., Denny et al., 2024), while noting that separating this study from a full HCI evaluation is a standard approach (Koedinger et al., 2013). For our pedagogical strategies (#12-14), we will ground each in the literature, from using multiple responses to diagnose ambiguity (Wang et al., 2023) to defining new assembly instructions as a standard practice in computer architecture education (Patt & Patel, 2003).
> > >
> > > We believe these revisions will substantially strengthen the paper, and we appreciate the opportunity to refine our work based on your insightful feedback.

---

> > > > ### Author Response · Authors · 2025-06-06
> > > >
> > > > (We will exceed the character limit above with references of additional papers we will cite, so we put it here.)
> > > >
> > > > American Educational Research Association, American Psychological Association, & National Council on Measurement in Education. 2014. Standards for Educational and Psychological Testing. American Educational Research Association.
> > > >
> > > > Brennan, K., and Resnick, M. 2012. New frameworks for studying and assessing the development of computational thinking. In Proceedings of the 2012 annual meeting of the American Educational Research Association.
> > > >
> > > > Denny, Paul, MacNeil S., Savelka J., Porter L., and Luxton-Reilly A. 2024. Desirable Characteristics for AI Teaching Assistants in Programming Education. In Proceedings of the 2024 on Innovation and Technology in Computer Science Education V. 1 (Turku, Finland), 408–414.
> > > >
> > > > Kafai, Y. B., and Resnick, M. (Eds.). 1996. Constructionism in practice: Designing, thinking, and learning in a digital world. Routledge.
> > > >
> > > > Kazemitabaar, M., et al. 2024. Codeaid: Evaluating a classroom deployment of an LLM-based programming assistant that balances student and educator needs. In Proceedings of the CHI Conference on Human Factors in Computing Systems.
> > > >
> > > > Koedinger, K. R., Brunskill, E., Baker, R. S. J. d., McLaughlin, E. A., & Stamper, J. 2013. New Potentials for Data-Driven Intelligent Tutoring System Development and Optimization. AI Magazine, 34(3): 21–34.
> > > >
> > > > Kumar, S. A. S., Yan, H., Perepa, S., Yue, M., & Yao, Z. 2025. Can LLMs Simulate Personas with Reversed Performance? A Benchmark for Counterfactual Instruction Following. arXiv preprint arXiv:2504.06460.
> > > >
> > > > Lye, S. Y., and Koh, J. H. L. 2014. Review on teaching and learning of computational thinking through programming: What is next for K-12? Computers in Human Behavior, 41(4): 51–61.
> > > >
> > > > Mannekote, A., Davies, A., Kang, J., & Boyer, K. E. 2024. Can LLMs Reliably Simulate Human Learner Actions? A Simulation Authoring Framework for Open-Ended Learning Environments. arXiv preprint arXiv:2410.02110.
> > > >
> > > > Papert, S., & Harel, I. 1991. Constructionism. Ablex Publishing Corporation.
> > > >
> > > > Patt, Y. N., & Patel, S. J. 2003. Introduction to Computing Systems: From Bits and Gates to C and Beyond. McGraw-Hill Higher Education.
> > > >
> > > > Zamfirescu-Pereira, J. D., Wong, R. Y., Hartmann, B., and Yang Q. 2023. Why Johnny Can’t Prompt: How Non-AI Experts Try (and Fail) to Design LLM Prompts. In Proceedings of the 2023 CHI Conference on Human Factors in Computing Systems (CHI '23), Article 437, 1–21.

---

### Author Response · Authors · 2025-06-02
**Summary of Changes**

We thank all reviewers for their insightful feedback. Integrating all your comments (you can find individual responses under your original comment), we will revise our manuscript to enhance clarity, rigor, and impact.

Key revisions include:

Overall Clarity & Detail: We will provide more detailed explanations and concrete examples of our question generation strategies, the Socrates system, and the pedagogical rationale for "Learning by Teaching."

Introduction & Motivation: The introduction will distinguish LLM use as a 'utility' versus for 'learning' CS. We will better motivate Section 7 (Evaluation) earlier, add explicit research questions, clarify vague statements (e.g., initial cost/answer quality claims, linking them to later data), and strengthen claims about pedagogical benefits (lines 41-45, 148-149) with citations on the "protégé effect." We will also
Related Works (Sec 3): This section will be significantly revised for more substantive descriptions of cited works (e.g., Brown & Garcia, Jeon & Lee), focusing on their primary contributions and relevance to our study, rather than superficial summaries. We will address how prior art informs challenges like educator misuse of LLMs. The "comprehensive" claim will be rephrased. All citations will be checked for completeness (e.g., missing years).

Methodology:

•	Question Design (Sec 4): We will restructure for better flow, provide more detailed explanations of our two strategies, and clarify how Strategy 1 (non-existing scenarios) necessitates student instruction to bridge LLM knowledge gaps, including concise in-text examples. We will discuss re-encoding's pedagogical effects, the risk of students "hacking" by remapping problems (and how even this involves learning), and revise line 167 for confident phrasing. Sec 4.2 (Case Study) will be more concise in the main text, with details moved to the appendix.

•	Prompt Engineering (Sec 5): We will clarify that guidance supports students effectively teaching the LLM, not general prompt engineering, and address "artificial prompt tweaking" by explaining how CoT/self-consistency make conceptually sound but imperfect instructions more understandable.

•	System Design (Sec 6): We will clarify that system requirements derive from pedagogical/practical needs. The lack of a GUI for question design (JSON used for flexibility/prototyping) will be noted as future work. Figure 1's simplicity will be contextualized (core contribution is pedagogy, not a novel NLP pipeline). Grader functionality (LLM for verification scalability, not replacing instructor oversight) and its non-CoT verification prompt will be elucidated.

Evaluation (Sec 7): We will enhance student outcome reporting with sample sizes for 'before'/'after' cohorts. We will incorporate statistical test results (means, SDs added to Fig 2, p-values) for performance improvements. Preliminary analysis shows significant improvement for 'Assignments' (p≈0.028) & 'Projects' (p≈0.018). This supports the "significant" claim (line 407). Non-significant results for 'Exams'/'Evaluations' will be discussed. Limitations (single course, historical data, no long-term retention) and pre-existing course conditions will be clearly described. Claims on LLM cost-effectiveness/capabilities will be carefully rephrased based on our specific data.

Discussion, Limitations, Future Work: We will expand on mitigating student over-reliance (our method makes students active instructors). We will discuss scalability across institutions/disciplines. Limitations (evaluation scale, usability, re-encoding effects) will be expanded. Future work, like exploring machine unlearning for LLM knowledge suppression (while noting our prompt-based approach prioritizes low cost/accessibility) and addressing student challenges with lower-capability LLMs (framing it as potentially pedagogically valuable), will be included.

Language, Background, Presentation: The manuscript will be thoroughly proofread. Redundant LLM/CoT basics will be condensed. BERT's description will be re-evaluated for conciseness/relevance.

---

### Decision · Program_Chairs · 2025-07-08

**Decision:**

Accept

**Comment:**

This paper proposes a highly original "Learning by Teaching" method, where CS students instruct LLMs. Despite initial clarity and methodological issues, the authors' significant revisions, including statistical analysis, promise a high-quality final version. This work is significant for AI in education, combating LLM over-reliance via active learning, showing promising preliminary results, and offering a valuable framework with the practical Socrates system.